# Exploring YouTube content creators' perspectives on generative AI in language learning: Insights through opinion mining and sentiment analysis

**Mazhar Bal[1], Ayşe Gül Kara Aydemir [2]\*, Mustafa Coşkun[3]**

**1** Faculty of Education, Department of Turkish and Social Sciences Education, Akdeniz University, Antalya, Turkey, **2** Department of Educational Sciences, Akdeniz University, Antalya, Turkey, **3** Bornova Science and Art Center, Izmir, Turkey

\* aysegulkara@akdeniz.edu.tr

**Data Availability Statement:** The data obtained from YouTube videos for the purpose of this study are openly available in https://osf.io/qy328/?view_only=a3b1b509673d4d72988f40bbae82a8e4. It is

## Abstract

This study aims to capture the stance of YouTube video content creators regarding the use of generative AI for language learning. Opinion mining and sentiment analysis techniques were employed to analyse the content, comments, and transcriptions of 66 YouTube videos published from December 2022 to October 2023. The findings revealed that most videos focused on speaking (n = 40) and writing skills (n = 24), with fewer videos addressing listening (n = 3) and reading (n = 19) skills. Sentiment analysis showed that videos predominantly conveyed optimistic (n = 42) and analytical (n = 17) sentiments, indicating a generally positive stance towards generative AI for language learning. Clustering analysis identified four thematic clusters: "language development and practices" (n = 33), "basic expression skills" (n = 25), "intercultural communication skills" (n = 6), and "language structure and meaning" (n = 2), representing different approaches to language learning with generative AI. Cross-sectional analyses revealed fluctuations in video counts and sentiment scores over time, with higher expectations for generative AI in writing and speaking skills, and relatively lower interest in listening skills. The findings suggest that YouTube video creators generally view generative AI as a promising tool for language learning, with a focus on developing practical communication skills, fostering intercultural understanding, and facilitating language development. These insights can inform the design and implementation of AI-supported language learning materials and practices.

## 1. Introduction

Learning now knows no boundaries, transcending the confines of formal educational institutions, presenting an expansive and borderless landscape for learning. The ubiquity and pervasiveness of technology empower individuals to gain knowledge at any place and time [1–3]. Language learning has become one of the prime areas that can benefit greatly from these advancements [4,5], and they attracted both learners and educators. As language and

also presented as supporting information. We confirm that our Supporting Information files do not contain any identifying data. Our study utilized transcripts of publicly available videos on YouTube, which are already accessible to the general public and do not contain any identifying information beyond what is already publicly shared by the content creators themselves. We have ensured that no indirect identifiers that could potentially compromise participant privacy are included in our data. In the data collection and analysis processes of this study, we adhered to YouTube Terms of Service, YouTube Developer Policy, YouTube API Policy, and Google Terms of Service, as well as relevant legislation. No personally identifiable information was used, in full compliance with the aforementioned terms of services.

**Funding:** The author(s) received no specific funding for this work.

**Competing interests:** The authors have declared that no competing interests exist.

technology are inextricably linked, there has been a proliferation of technologies in the field of language learning, and each new technology has had its own place. In the literature of technology enhanced language learning (TELL), computers have notably played an important role [6,7], providing learners with digital platforms for language acquisition. The advent of mobile technologies has further enriched the learning experience [8], as evidenced by studies conducted by [9–11]. Immersive technologies including augmented, virtual, and mixed realities, have emerged as powerful tools in language education. Studies showcase the impact of these technologies on creating immersive language learning environments [12–18]. Furthermore, Web 2.0 tools providing interactive and collaborative learning experiences, fostering dynamic and participatory learning environments [19,20] have had greatly influenced language learning and teaching. A review study of 398 articles on technology-enhanced language learning, finding learners had positive perceptions. Their systematic analysis showed few negative results, suggesting technology's favorable reception in education, potentially yielding positive language learning outcomes [21].

As the integration of various technologies in language education continues at a rapid pace, a new frontier has emerged that promises to revolutionize the field even further. Recently, generative AI technologies have emerged as game-changers, introducing innovative approaches across various domains of life, with language learning being no exception. Generative AI is a term used to describe machine learning solutions that are trained on large amounts of data to generate high-quality content, including text, code, images, audio, and other types of media [22–24]. Throughout the history, it was believed that artistic and creative works, such as writing poetry, developing software, designing fashion, and composing music, were exclusively human activities [25]. However, this has changed with the advent of generative AI, which is now capable of performing these tasks with remarkable proficiency.

Advancements in natural language processing and deep learning have led to the development of large language models (LLMs) [26], which have reached a level of maturity that allows them to be effectively integrated into nearly all aspects of daily life. While interest in generative AI has rapidly increased since November 2022, significant progress has been already made in the development process of this technology and advancements have occurred in various aspects. Google's BERT (Bidirectional Encoder Representations from Transformers), which was introduced in 2018, has had a transformative impact on natural language processing. It has enabled more accurate and efficient understanding of context and meaning in text, as stated by Devlin et al. [27]. OpenAI's ChatGPT, developed based on the GPT-3.5 architecture, was released in November 2022 and has since attracted worldwide attention for its remarkable capabilities [28]. It took only 5 days for ChatGPT to reach 1 million users, and it has shown an extraordinary increase in the number of users, reaching 200 million users as of March 2024 [29]. Beyond Chat GPT, numerous other generative AI models are now in use, serving a diverse range of users. In December 2023 Google Deepmind released "Google Gemini", a multimodal generative AI tool leveraging the power of multi-model generative AI technology [30,31].

Other notable generative AI tools include OpenAI's DALL-E, which is designed to generate high-quality images from textual descriptions, by combining advanced natural language processing and computer vision techniques, showcasing the potential of AI in creative fields [32,33]. Microsoft's GitHub Copilot is another noteworthy tool that aids developers by offering smart code suggestions and automating repetitive coding tasks, thus boosting productivity and fostering innovation in software development [34]. Additionally, OpenAI's Codex, which powers GitHub Copilot, extends its capabilities to more complex programming tasks and supports multiple programming languages [35]. Moreover, tools like Jasper AI are transforming the content creation industry by enabling automated writing of marketing copy, articles, and

social media posts with human-like quality [36]. Soundful, Beatoven.ai, and Google's MusicLM, other innovative tools, generates music based on textual descriptions [37]. Adobe's Sensei leverages AI to improve creative processes by providing features such as automated photo editing, intelligent tagging, and content-aware fill. These tools exemplify the diverse applications of generative AI, and it is apparent that generative AI technology have very high potential to transform the way we produce knowledge, do business, learn, make decisions, and utmost the way we live.

As we examine the potential for generative AI to transform various domains, its application in education, particularly language learning, emerges as a promising area of exploration. Generative AI technology holds the promise of further enhancing the efficiency and effectiveness of language learning experiences, offering innovative approaches and personalized learning opportunities for learners in the ever-evolving landscape of education [38]. They have potential to serve as a learning assistant such as chatbots in language learning including first language learning, second language learning, and foreign language learning [39]. Among the aforementioned tools, ChatGPT is particularly noteworthy for its distinctive capabilities and extensive adoption, largely attributable to the complimentary features it offers, which have significantly contributed to its considerable user base. The concept of "search is learning" was introduced with the release of ChatGPT (Generative Pre-trained Transformer) in November 2022 [40]. ChatGPT is an AI-powered chatbot that mimics a virtual robot that can engage in human-like conversations [41]. Unlike conventional AI models, ChatGPT is able to retain context even with increasing inputs, resulting in more relevant and accurate responses [42]. In this study it is noted that this capability ensures that learners can engage in a seamless and uninterrupted natural learning environment. This feature helps learners participate in a smooth and uninterrupted natural learning environment. This innovative approach underscores the transformative role that large language models like ChatGPT could play in redefining the dynamics of learning, where the act of seeking information becomes an integral part of the learning process, fostering a more interactive and engaging learning experience. Setting from this point, it is essential to explore how generative AI technologies such as ChatGPT could be further utilized in language learning.

## 1.1 Generative Aartificial intelligence in language learning

As the demand for language learning grows across sectors [43], the focus shifts to leveraging new technologies for the benefit of learners and instructors. Generative AI is gaining popularity among language learners and educators. Generative AI tools can significantly enhance language learning by providing essential elements such as suitable content, ample practice opportunities, progress tracking, and effective feedback [44]. Reactions to the use of generative AI have ranged from optimism to fear regarding the future of education [45,46]. Addressing this hot issue, numerous studies have been conducted, focusing on exploring the potential and challenges of generative AI in enhancing language learning experiences. In the context of language learning, generative AI technologies like ChatGPT has various applications, including text generation, machine translation, content creation, grammar correction, and responding to questions [42]. They noted that ChatGPT is a powerful tool that creates customized learning content in over 12 languages and customizes the output based on user preferences, providing a flexible and personalized learning experience. It is argued that by engaging in interactive dialogues and contextual exercises, ChatGPT provides a secure and supportive environment for practicing language skills. This allows for natural practice in text-based conversations and writing, similar to interacting with a native speaker [47]. Accordingly, [48] contended that it is possible to benefit from ChatGPT in relation to language learning as long as activation of the

basic language skills have been supported. They expressed that language learning requires practice, repetition; however, all of which are lacking in ChatGPT.

In their quasi-experimental study involving 120 Thai students aged 19–20 from the first year of preservice teachers, Songsiengchai et al. [49] present a case for integrating ChatGPT into English language learning curricula. Their findings demonstrate that artificial intelligence (AI) tools such as ChatGPT can significantly enhance Thai students' English language proficiency, a consistent improvement in language skills was observed. In the follow-up interviews, students in the experimental group reported a more engaging and personalized learning experience. The authors emphasized the impact of real-time feedback and interactive exercises provided by ChatGPT. Another study, conducted by Li et al. [50], demonstrated the potential of ChatGPT for personalized learning in the process of language learning. They conducted an exploratory analysis to examine the language learning experiences of YouTube content creators and the role of Chat GPT in their self-directed learning. Interviews were conducted with 19 YouTubers. The authors recognize that ChatGPT can be utilized as a resource to enable students to investigate topics, seek clarification, and engage in critical thinking, which fosters a more learner-centered and personalized learning experience.

A content analysis of 140 YouTube videos was conducted by [51] to explore language learning community perceptions. They found ChatGPT to be a valuable tool for language learning. Additionally, [52] examined YouTube content creators' views on ChatGPT for language learning. They highlighted its benefits while also addressing concerns such as bias, misinformation, and past failures of novel technologies in language learning. The sudden availability of applications like ChatGPT from OpenAI in late November 2022 suggests the possibility of automatic text generation in human-like quality. While initially celebrated for its impressive outputs, scepticism emerged regarding its flaws and errors. Despite its recent unveiling, studies on ChatGPT's role in language learning are limited [42]. As AI-powered tools become commonplace, language teachers and learners must develop advanced digital competencies [53] to maximize benefits and navigate potential risks effectively.

The aforementioned studies demonstrate that generative AI tools, particularly ChatGPT, have significantly enhanced language learning by offering personalized and interactive learning experiences. Such tools provide essential elements, including suitable content, multiple practice opportunities, real-time feedback, and progress tracking. Research has shown that ChatGPT has the potential to create personalized learning materials in various languages. This enables students to engage in interactive dialogues and contextual exercises, which fosters a secure and supportive environment for practicing language skills. The use of ChatGPT has shown promising results in improving language skills and has been recognized for its ability to make learning more engaging and personalized. Nevertheless, challenges such as the necessity for practice and repetition, as well as concerns about bias and misinformation, underscore areas that require further research and development. Despite the growing popularity and promising potential of generative AI tools like ChatGPT in language learning, there remains a significant research gap in understanding their full range of applications and efficacy in language learning.

One of the most commonly accepted and frequently heard expressions nowadays is "You can learn anything from You Tube". Individuals from diverse backgrounds contribute a wide range of content on YouTube, providing a unique richness and breadth of material for audiences worldwide. The platform serves as a melting pot for people from all walks of life [54]. Whether it's mastering a new skill, delving into academic subjects, or exploring diverse perspectives, YouTube has evolved into a dynamic and inclusive space. It is also considered to be an invaluable space for language learners.

## 1.2 Purpose of the study and research questions

The present study aims to capture the stance of YouTube video content regarding the use of generative AI for language learning. It seeks to provide insight into which language skills (reading, writing, speaking, and listening) are addressed and how YouTube videos are clustered. The research questions that guided this study and the research question statements are as follows:

**RQ1:** How do YouTube videos about generative AI in language learning specifically address different language skills such as speaking, listening, reading, and writing?

This research question examines how generative AI is addressed in YouTube videos related to language learning, with a particular focus on language skills such as speaking, listening, reading, and writing.

**RQ2:** What is the distribution of sentiments (Optimistic, Distrustful, Mixed, Analytical, Ethical, Biased, Futuristic, Neutral) expressed in YouTube videos about the use of generative AI for language learning?

This research question aims to analyze the distribution of sentiments (Optimistic, Distrustful, Mixed, Analytical, Ethical, Biased, Futuristic, Neutral) expressed in YouTube videos about using generative AI for language learning. This helps us understand the YouTube content creators' stance towards generative AI in language learning.

**RQ3:** How are YouTube video contents clustered in terms of the wording by using text mining methodologies?

This research question aims to identify the vocabulary and phrases that are commonly used in YouTube videos about generative AI and language learning. This will be achieved through the application of text mining methodologies. This analysis will provide insights into the themes around which the content is centered and reveal whether certain word groups are frequently used, thereby offering information about content types and focal points.

**RQ4:** What are the results of cross-sectional evaluations of the clustering, classification, and sentiment analyses of YouTube video contents about generative AI in language learning?

This research question aims to examine the results of clustering, classification, and sentiment analysis. It evaluates how videos are categorized by content and sentiment and what these categories mean. The findings will highlight prominent content types and sentiments in language learning with generative AI.

## 2. Method

This exploratory study employs text mining and sentiment analysis techniques to find out how YouTube video contents relate generative AI and language learning to language skills, and capture the stance of YouTube content creators on how generative AI could be used for language learning. Text mining refers to the process of extracting knowledge which is necessary directly for making the important decision from textual data [55].

Ethics approval was not required for this study as it did not involve primary data collection from human subjects. The data were obtained from YouTube videos, where information is freely available in the public domain. The YouTube API was utilized to access videos on topics related to Generative AI and/or ChatGPT and language learning. In the data collection and analysis processes of this study, we adhered to YouTube Terms of Service [56], YouTube Developer Policy [57], YouTube API Policy [58], and Google Terms of Service [59], as well as relevant legislation. No personally identifiable information was used, in full compliance with the aforementioned terms of services. This study recognizes the high prevalence and widespread acceptance of ChatGPT, along with other generative AI tools, globally. As a result,

"ChatGPT" has been included as a distinct keyword in our search queries. The following search query was used for the YouTube API:

querystring = 'intitle:("generative AI" | "ChatGPT") & intitle:("language learning")'

The query was executed using the "googleapiclient.discovery" Python library, which is a tool for interacting with YouTube API. The initial phase of the search yielded a dataset comprising 92 YouTube videos that were identified through the specified search query. Following the acquisition of this initial dataset, a filtration process was then applied by the researchers to assess their relevance to the main focus of the study. This filtration involved a systematic and criteria-driven assessment to ensure that only content directly related to generative AI in language learning was retained for further analysis. The researchers evaluated the video's alignment with the study's objectives by considering factors such as its title, description, and tags. They have also reviewed the video's content to assess the accuracy and depth of the information presented. The purpose of this filtration process was to refine the dataset, ensuring that subsequent analysis is based on a focused and relevant collection of YouTube videos that make a meaningful contribution to the study of generative AI in language learning.

Out of the total, 26 videos were excluded based on specific criteria as below:

- There are 3 videos with English titles, but the content is presented in Chinese.

- 1 video has a title in English; however, the content is presented in Hindi.

- There are 10 videos that are classified as very brief, with each one lasting between 0–15 seconds.

- 3 videos whose titles aligned but upon analysis, their content was found to be unrelated.

- 1 duplicate video.

- 6 videos that appeared in the YouTube API search results are not currently accessible.

- 2 videos featured instrumental music without any spoken content.

The remaining 66 videos were subjected to further analysis in line with the purpose of the current study. After accessing and filtering videos, video comments were collected using a relevant API. Furthermore, video transcriptions were gathered for subsequent text analysis, utilizing the "youtube_transcript_api", a Python library designed to retrieve transcriptions directly from YouTube videos. The addition of video transcriptions enhances the analysis by enabling researchers to examine the actual spoken or written content within the videos. This comprehensive approach, which includes video content, and transcriptions, allows for a thorough exploration of the discourse surrounding generative AI in language learning on the YouTube platform.

Subsequently, the transcriptions obtained from YouTube videos underwent classification using the ChatCompletion class of the OpenAI library, employing the GPT-4-0613 model. The accuracy level of GPT-4 is mentioned as high in the Capabilities Declaration research page of the official OpenAI website [60]. The aim of this classification was to categorize these transcriptions into classes with a focus on key language skills such as reading, writing, listening, and speaking. The prompt used for the GPT-4-0613 model was structured as follows:

'Respond in the JSON format for the following YouTube transcription about ChatGPT in a Language Learning Video: {"about": category_name}, where the possible category_name categories are Reading, Writing, Listening, Speaking'

In the initial classification, the researchers who analyzed the results found by the GPT-4-0613 model observed that some transcriptions were erroneously categorized as "speaking" due to their nature as video speech transcripts. Therefore, the classifications were manually revised

by two subject matter experts to rectify the misclassification and ensure the accuracy of the categorization process. To ensure the reliability of our coding process, we conducted an interrater reliability analysis using Cohen's kappa. This analysis involved 66 video transcripts, independently coded by two researchers into four categories: reading, writing, speaking, and listening.

We identified discrepancies in 9 out of the 66 video transcripts: 1 for listening, 1 for reading, 5 for speaking, and 2 for writing.

The observed agreement (P_o) was calculated as follows:

$P_o = 57/66 = 0.8636$

The expected agreement (P_e), assuming equal distribution across categories, was:

$P_e = (57/66)^2 + (57/66)^2 + (57/66)^2 + (57/66)^2 = 0.25$

Cohen's kappa ($\kappa$) was then calculated:

$\kappa = (P_o - P_e)/(1 - P_e) = (0.8636 - 0.25)/(1 - 0.25) = 0.818$

The kappa value of 0.818 indicates almost perfect agreement [61] demonstrating strong interrater reliability and supporting the validity of our current process.

The categorization of opinions expressed in the transcripts about ChatGPT was performed using the GPT-4-0613 model. The researchers did not rely solely on a single source for the classes. Initially they analyzed the prime source on sentiment analysis [62], and also reviewed additional sources on YouTube sentiment analysis [63–65]. Based on this comprehensive review, the researchers selected the classes that are mutually exclusive and collectively exhaustive for extracting sentiments. The model was utilized to structure results for categorical opinion mining analysis, with the aid of a specified prompt. The prompt used for this operation was:

'Respond in the JSON format for the following Youtube transcription about ChatGPT in Language Learning Video: {"sentiment": sentiment_classification1, sentiment_classification2 (if needed) and sentiment_score: (-1 to 1)}, where the possible sentiment_classification categories are Optimistic, Distrustful, Mixed, Analytical, Ethical, Biased, Futuristic, Neutral'.

The results obtained from this process have been saved for further analysis, with the aim of understanding and categorizing the opinions expressed about ChatGPT in the context of language learning videos. As stated in the prompt, the GPT-4-0613 model was tasked with conducting sentiment analysis on the transcripts, without any subjective assessments. In addition to the aforementioned analyses, a clustering analysis was conducted using Latent Dirichlet Allocation (LDA). LDA is a topic modelling technique used to identify topics within a collection of text documents [66]. The transcripts underwent tokenization, followed by the preparation of token sets through the removal of punctuations and stop words. A silhouette score analysis was utilized to determine the optimal number of clusters. Fig 1 shows the silhouette scores.

This analysis assesses the consistency and distinctness of clusters by testing various cluster numbers and selecting the number that maximizes both intra-cluster cohesion and inter-cluster separation [67]. Upon conducting the silhouette score analysis, it was observed that the dataset exhibited the highest coherence when clustered into four distinct groups. Two subject matter experts carefully examined and labelled these clusters to ensure a comprehensive understanding and categorization. The experts assigned descriptive labels to the clusters, resulting in the following categories: "Basic Expression Skills", "Intercultural Communication Skills", "Language Development and Practice", and "Language Structure and Meaning". These labels offer a comprehensive overview of the identified thematic clusters within the analyzed text dataset.

In summary, this study utilized the YouTube API to access videos related to generative AI, ChatGPT, and language learning. A filtering process was carried out to ensure relevance, resulting in the analysis of 66 videos. Comments and video transcriptions were collected and categorized using both the OpenAI GPT-4-0613 model and sentiment classification prompts

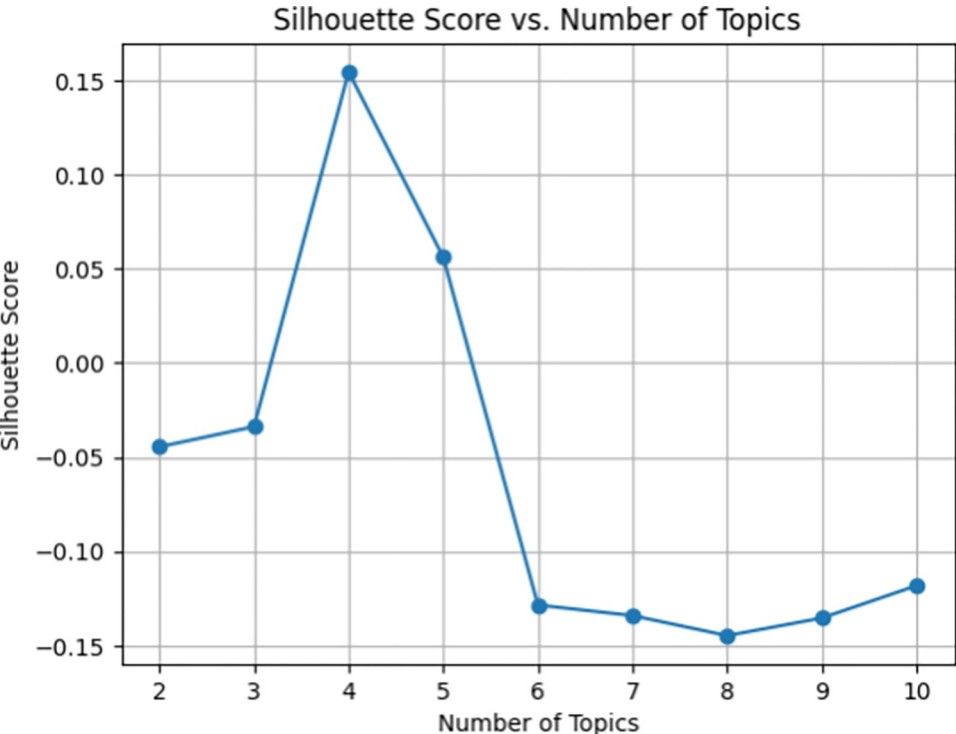

**Fig 1. Silhouette score by number of topics.**

to determine opinions about ChatGPT. Additionally, a clustering analysis using LDA was performed on the transcript data. The study employed various techniques such as tokenization, removal of punctuations and stop words, and silhouette score analysis to identify four distinct thematic clusters. These clusters were then labelled by subject experts. Code of sentiment analysis is presented in Appendix A.

Fig 2 presents the flow of method.

The findings gathered from these comprehensive analyses are presented and analyzed in the following sections.

## 3. Results

This study delves into YouTube video contents on the use of generative AI in language learning, examining their impact on basic language skills. Additionally, we analyze the sentiments of video contents towards generative AI for language learning. Utilizing text mining methodologies, we cluster video contents based on their wordings, revealing prevalent themes and patterns. Cross-sectional evaluations explore the results of clustering, classification, and sentiment analyses. The following section presents the findings of this study, organized according to each research question.

### 3.1 How do YouTube videos about generative AI in language learning specifically address different language skills such as speaking, listening, reading, and writing?

The study analyzed the YouTube video contents using the GPT-4-0613 model to categorize them based on basic language skills as listening, reading, writing, and speaking. Distribution of

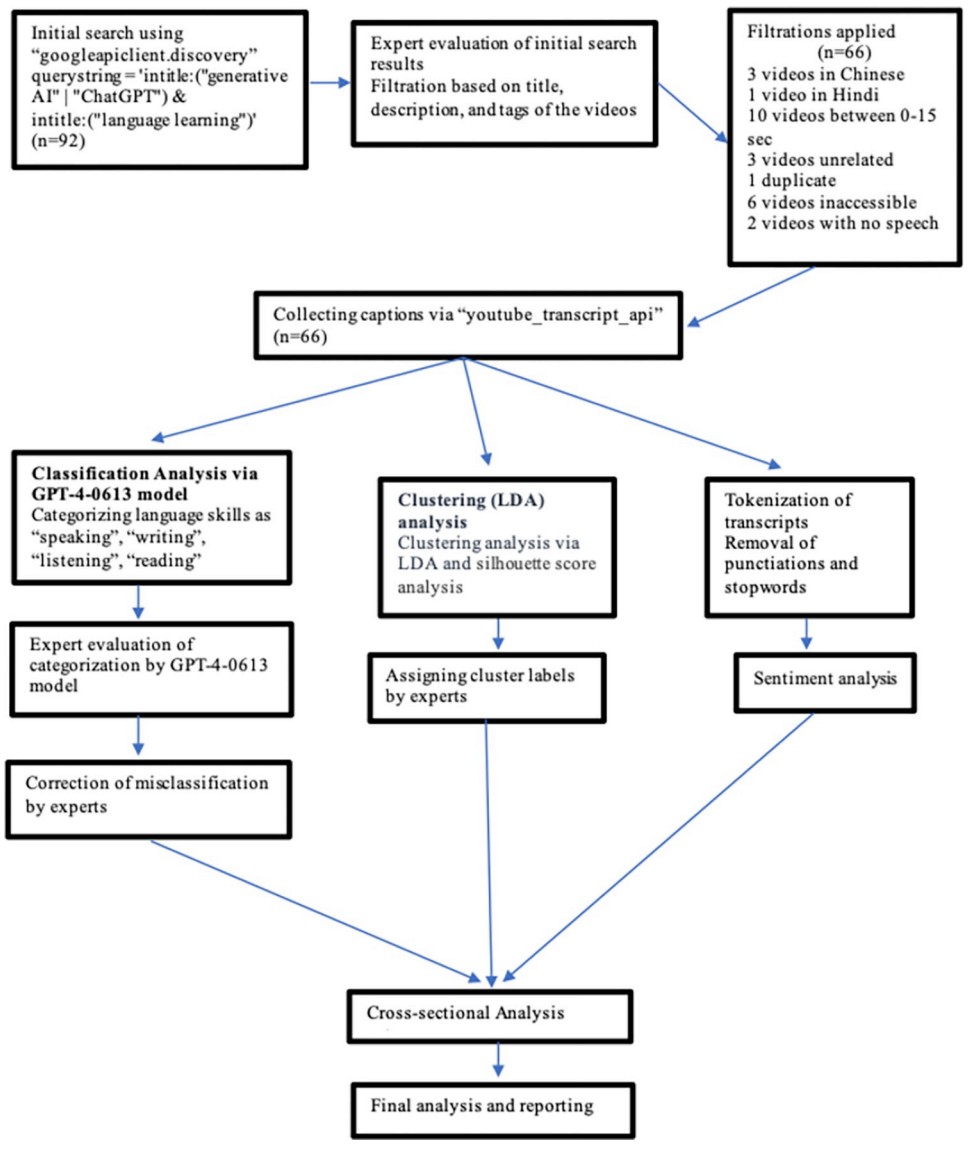

**Fig 2. Flow of method.**

YouTube video contents by language learning categories is explained. Fig 3 shows the categorization of the 66 included videos based on the categories.

Examining Fig 3, it is obvious that the total video count in Fig 3 is not 66, but rather 86. The reason behind this issue is that the initial analysis of 66 videos was conducted using GPT-4-0613 model. The results based on the analysis of GPT-4-0613 model yielded listening ($n = 4$), reading ($n = 5$), speaking ($n = 54$), and writing ($n = 3$) video counts, in total $N = 66$. In the second phase, two researchers reviewed the results provided by the GPT-0613 model. As a result of the researchers' analysis, it was observed that 30 videos were erroneously miscategorized by GPT-4-0613. The researchers conducted a thorough review of the video transcripts and found that some of the videos addressed more than one language learning skills, requiring assignment to multiple categories. Consequently, the analysis of video content distribution by language learning skills category resulted in 86 video counts, as shown in Fig 3.

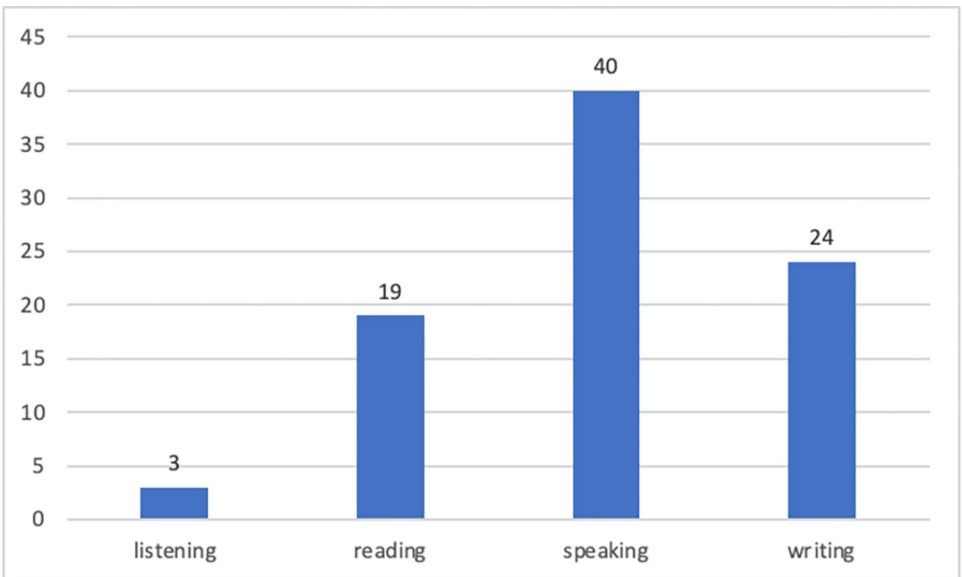

**Fig 3. Distribution of YouTube video counts by language skill categories.**

Most language learning videos prioritize speaking and writing, aiming to enhance practical communication skills. However, listening and reading receive less emphasis. Word clouds presented in Fig 4 highlights the words that stand out for each language learning skill category.

Fig 4 illustrates that videos on using generative AI in language learning cover a broad range of language content, guiding learners through various topics in basic language skill areas. The prominent words in writing skill suggest generative AI's ability to offer interactive learning experiences, aiding learners in improving language skills and creative writing. Similarly, the words emphasized in speaking skill indicate diverse content aimed at instructing learners on appropriate language use in different speech situations. In reading skills, the featured words reflect generative AI's aim to assist learners in understanding word meanings, interpreting texts, and comprehending multilingual content. Lastly, the emphasized words in listening skills suggest the diverse content available to enhance different language skills for learners, highlighting generative AI's potential to provide interactive and varied listening experiences to improve language skills.

### 3.2 What is the distribution of sentiments (optimistic, distrustful, mixed, analytical, ethical, biased, futuristic, neutral) expressed in YouTube videos about the use of generative AI for language learning?

This section presents the findings for sentiment classification and polarity of YouTube video contents towards generative AI for language learning, and comments counts of YouTube videos. The data for sentiment analysis is displayed in Fig 5.

Based on our analysis, it was observed that an optimistic sentiment was prevalent in the majority of the videos, with 42 out of 66 reflecting this trend, as shown in Fig 5. The prevalence of an optimistic perspective may be an indication of high hopes for generative AI for language acquisition. The analytical sentiment was observed in 17 videos, making it the second most frequently observed sentiment. Highly responsive YouTubers promptly produced content on how generative AI could be utilized for language learning once generative AI had become very popular, and it is not surprising that they adopted an analytical stance towards a technology

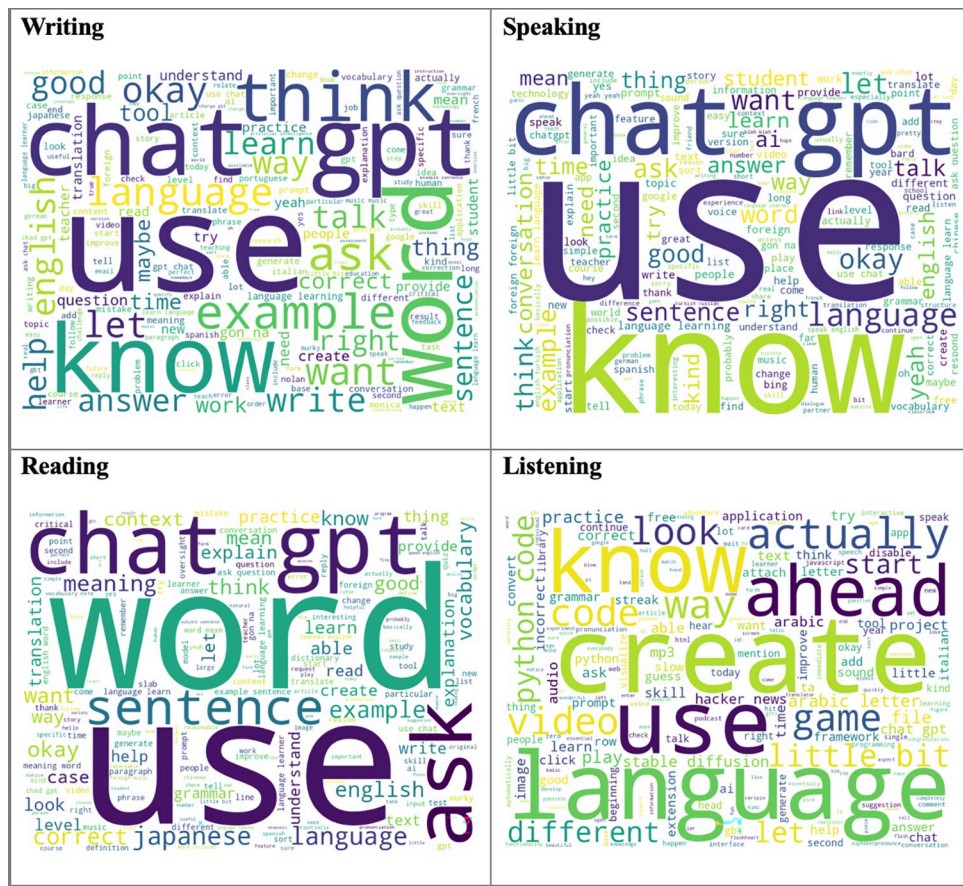

**Fig 4. Word cloud for basic language skill category.**

they were still exploring. Based on the findings, 3 videos were associated with mixed sentiments and 3 with neutral sentiment, indicating diverse sentiments towards the use of generative AI. Only one video reflects a distrustful sentiment, titled "ChatGPT just killed language learning" (https://youtu.be/gVpDSEBz9iw?si=3ogmVnu8f4NevUTi), presented by a polyglot who underscores the significance of social skills in language learning.

Additionally, GPT-4-0613 was used to calculate sentiment polarity scores, which quantitatively define the direction of the expressed sentiment, falling within a scale from -1.0 (indicating a negative sentiment) to +1.0 (representing a positive sentiment). Sentiment polarity scores of YouTube video transcripts by the number of videos were presented in Fig 6.

As seen in Fig 6, the sentiment polarity scores of YouTube video transcripts reveal that a significant portion of videos ($n$ = 54) have very high positive sentiment scores, with 38 of them falling within the range of 0.9 to 1.0, and 16 videos ranging from 0.8 to 0.9. Setting from this point, we could argue that majority of the videos view the use of generative AI for language learning positively. The remaining 11 videos, while having slightly lower scores, still exhibit a positive polarity. Only one video has a negative sentiment score of -0.7. This score belonged to the video titled "ChatGPT just killed language learning", and its sentiment classification were "distrustful", and it is expected to have a negative polarity due to its distrustful nature.

In addition, comment counts of the videos were given in Fig 7. The first video with 35 comments is https://www.youtube.com/watch?v=l41hZLRsDos created by Tom Gally and published just 1 week after the release of ChatGPT. The second one is (https://www.youtube.com/

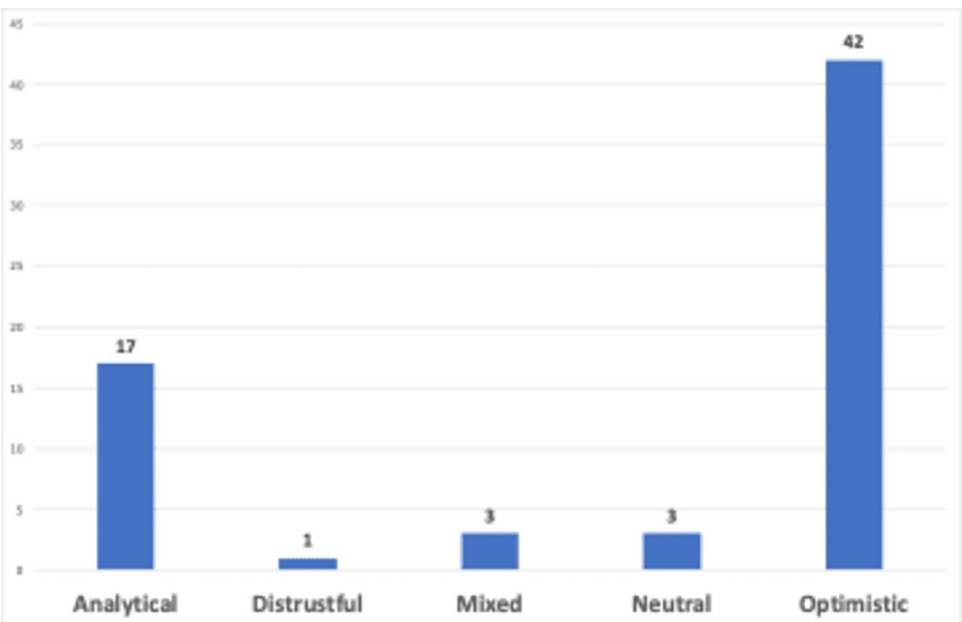

**Fig 5. Distribution of sentiment analysis.**

watch?v=kXChJfHZrjU) created by Jerry Registre and very argumentative/provoking about comparison of generative AIs and finding the best for language learning. Both videos received almost 100 comments while researchers reported on the current research. The remaining videos received less interaction, possibly due to the novelty of the topic and a lower level of reaction from the audience.

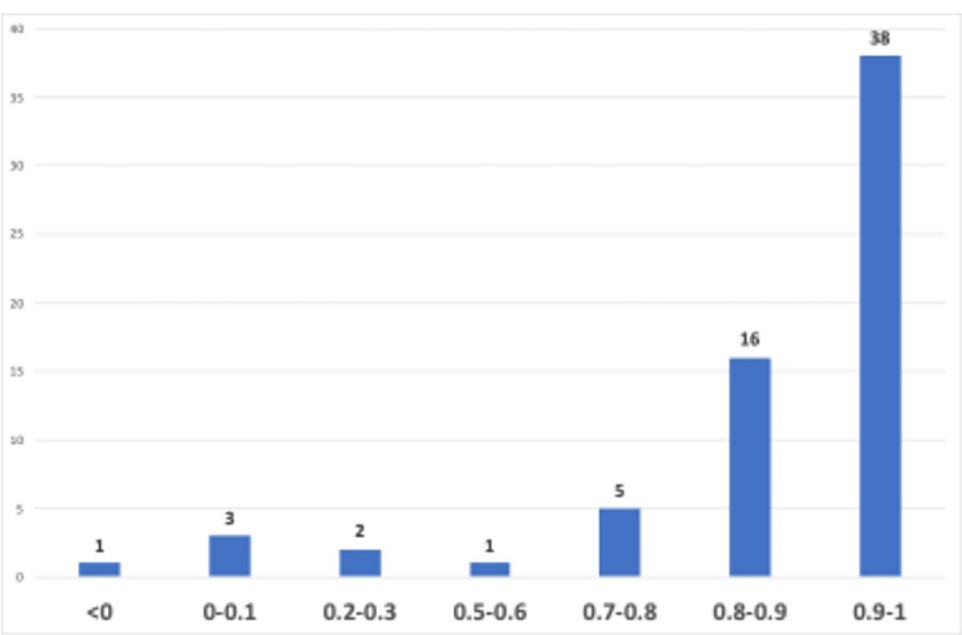

**Fig 6. Sentiment polarity scores.**

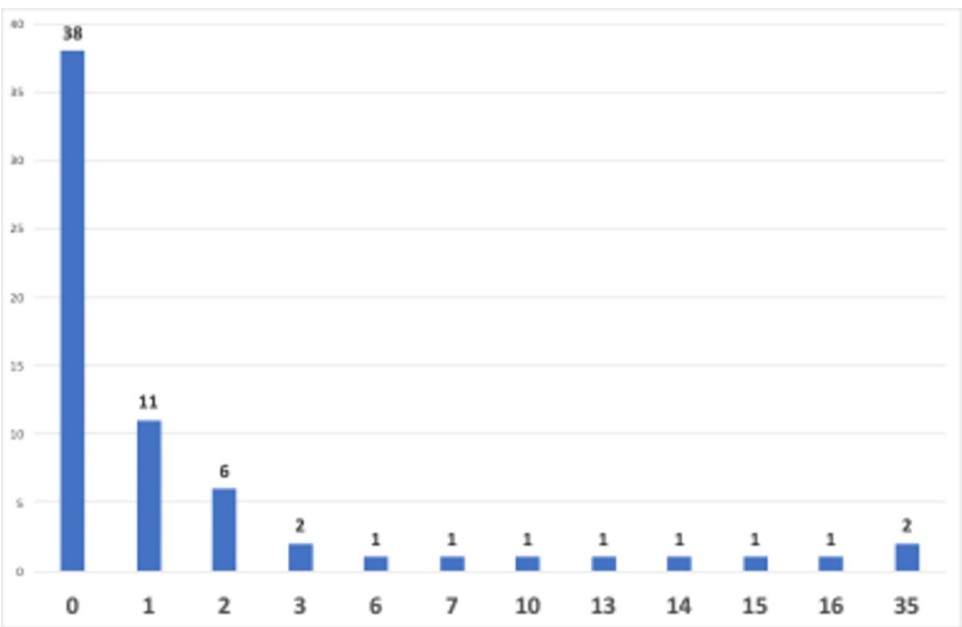

**Fig 7. Comment counts by video counts.**

### 3.3 How are YouTube video contents clustered in terms of the wording by using text mining methodologies?

Researchers identified four clusters by employing topic analysis technique to video contents as "language development and practices", "basic expression skills", "intercultural communication skills", and "language structure and meaning". Fig 8 shows the video contents' distribution of clusters by video counts.

The analysis in Fig 8 classifies videos into four clusters in terms of using generative AI in language learning: "language development and practices" (n = 33), potentially covering vocabulary expansion and real-life language use; "basic expression skills" (n = 25), likely focusing on expression forms; "intercultural communication skills" (n = 6), guiding language learners in interacting with different cultures; and "language structure and meaning" (n = 2), probably delving into in-depth language analysis.

### 3.4 What are the results of cross-sectional evaluations of the clustering, classification, and sentiment analyses of YouTube video contents about generative AI in language learning?

The Fig 9 displays the variation in video counts for the four main categories of language skills on a monthly basis. Videos addressing writing skills show an expected trend similar to "innovation trigger" phase of Gartner's hype cycle, and then shows an increasing interest as an indicator of high expectations, then followed by decrease. The trend in writing skills may be attributed to the fact that video creators have a general understanding of how generative AIs can enhance writing skills. The graphic displays video counts and publishing trends, which may indicate video creators' high expectations for generative AIs to improve their skills. The trend in speaking skills also suggests ongoing high expectations for this technology in acquiring speaking skills. The graphic displays fluctuations in reading skills, despite an increase in interest until March 2023. Based on the graphic and video counts related to listening skills, it

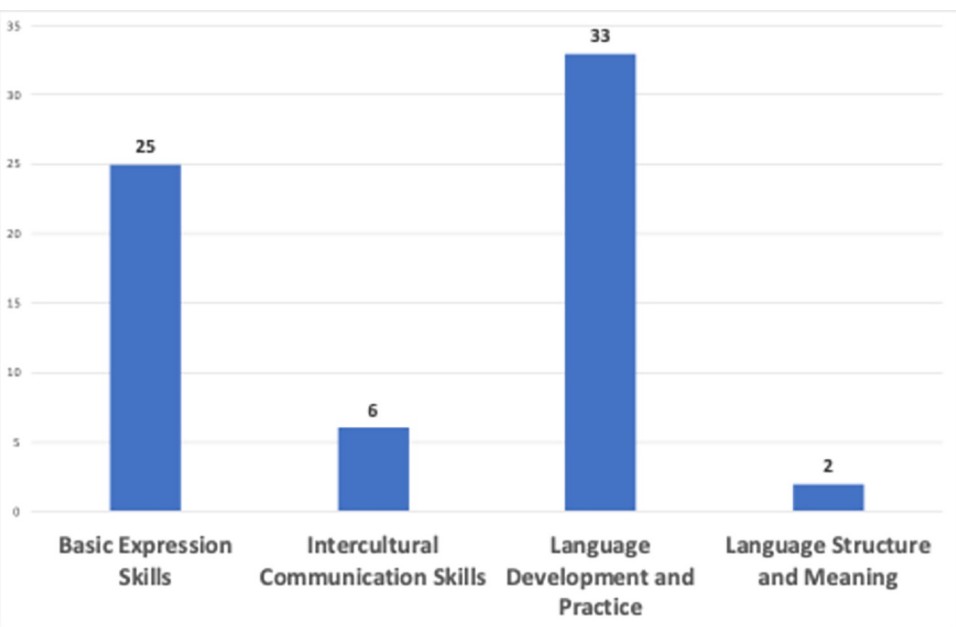

**Fig 8. Distribution of clusters.**

can be concluded that video creators are still uncertain about how to effectively utilize these tools for improving listening skills.

Upon examining the sentiment classes of videos by month, it becomes clear that the prevalent sentiment regarding the use of generative AIs in language learning is mostly optimistic, followed by analytical (Fig 10). Therefore, video creators mostly adopted an optimistic stance towards generative AIs for language learning, which could be attributed to their high expectations. The prevalence of an analytical stance is not surprising, given that generative AIs are still

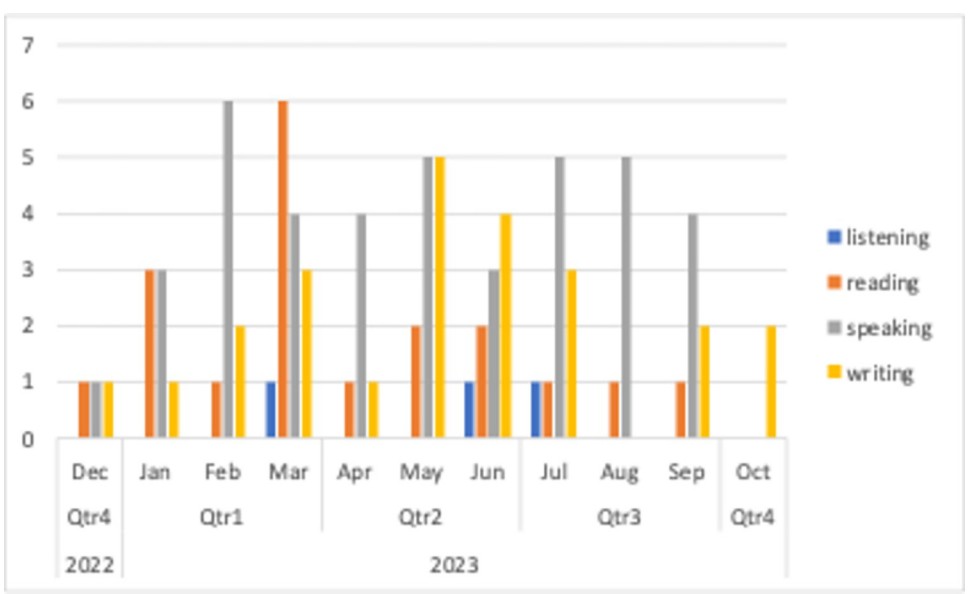

**Fig 9. Video counts of language learning categories by months.**

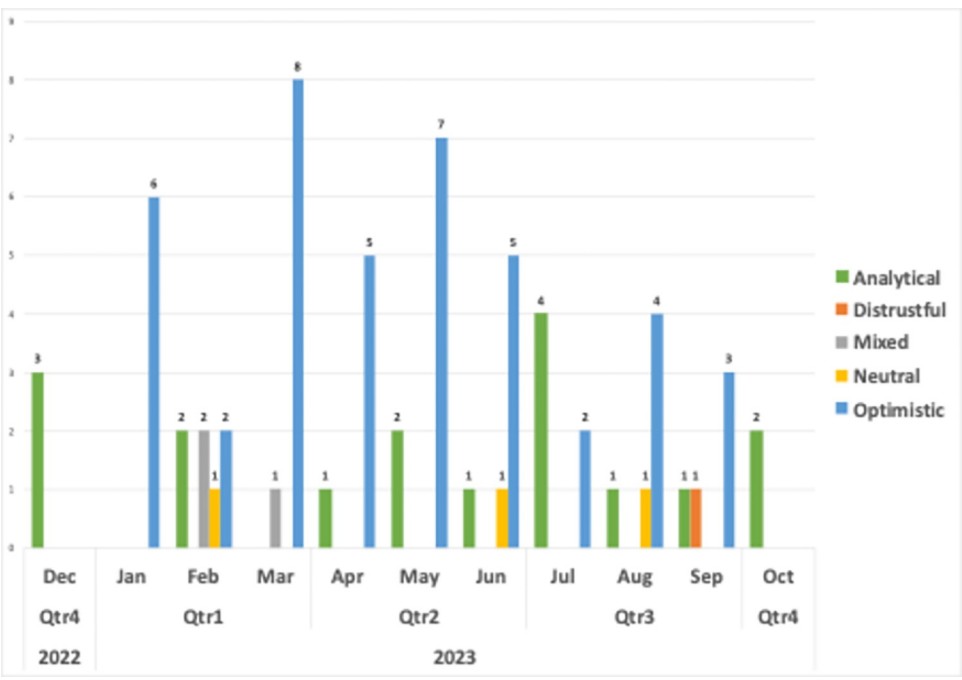

**Fig 10. Distribution of sentiment classes by months.**

in the exploration phase. Although artificial intelligence has a relatively long history in language learning, the affordances and capabilities of generative AIs are novel to most people. This result could show that video creators maintained a balanced perspective and avoiding biased language. Mixed sentiments disappeared in the first quartile. Furthermore, very limited expression of distrustful sentiment is consistent with our findings. The video titled "ChatGPT killed Language Learning" was found to contain only one expression distrustful sentiment, as previously described.

Fig 11 displays the average sentiment scores of the analyzed videos. It is worth noting that the video creators expressed predominantly positive sentiments with a slight fluctuation towards the use of generative AIs in language learning over the studied time period (from December 2022 to October 2023).

Fig 12 displays the average sentiment scores for YouTube videos related to the use of generative AIs in basic language skills. The sentiment scores for reading, speaking, and writing skills are highly positive (0.79, 0.78, and 0.78, respectively). Based on this finding it could be concluded that video creators embraced this technology with promising prospects for enhancing the aforementioned skills. The limited number of videos ($n$ = 3) addressing listening skills may lead to a misleadingly high positive average sentiment score.

Fig 13 shows the average sentiment score for each sentiment class. The average sentiment score for the "analytical" sentiment is 0.78, indicating a positive polarity. This result is consistent with the sentiment classification results. It is important to note that the sentiment score of 'Analytical' does not necessarily yield a positive polarity. Therefore, it is observed that video creators with analytical sentiments have positive expectations for generative AIs in language learning. Similarly, video creators with mixed sentiments also have positive expectations, as indicated by their average sentiment score of 0.30. The negative sentiment score of "distrustful" with -0.70 average sentiments score could be evidence for a valid analysis. A high average

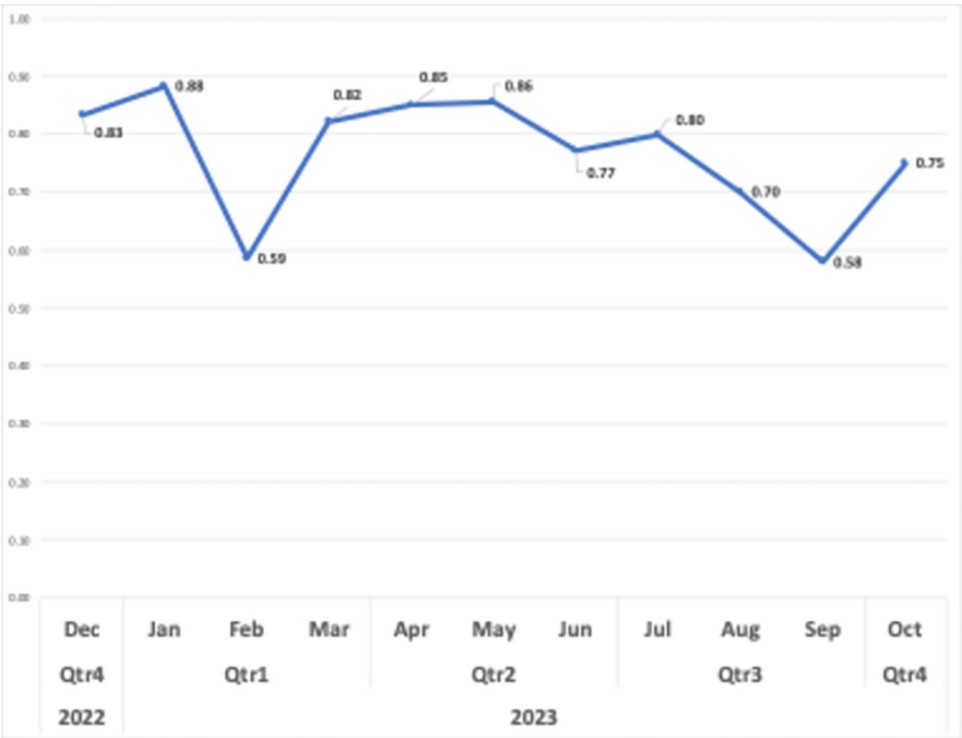

**Fig 11. Average sentiment scores.**

sentiment score of 'optimistic' could be associated with high hopes. The sentiment score is 'neutral' with a value of 0.0, as expected.

Cluster average sentiment scores are represented in Fig 14.

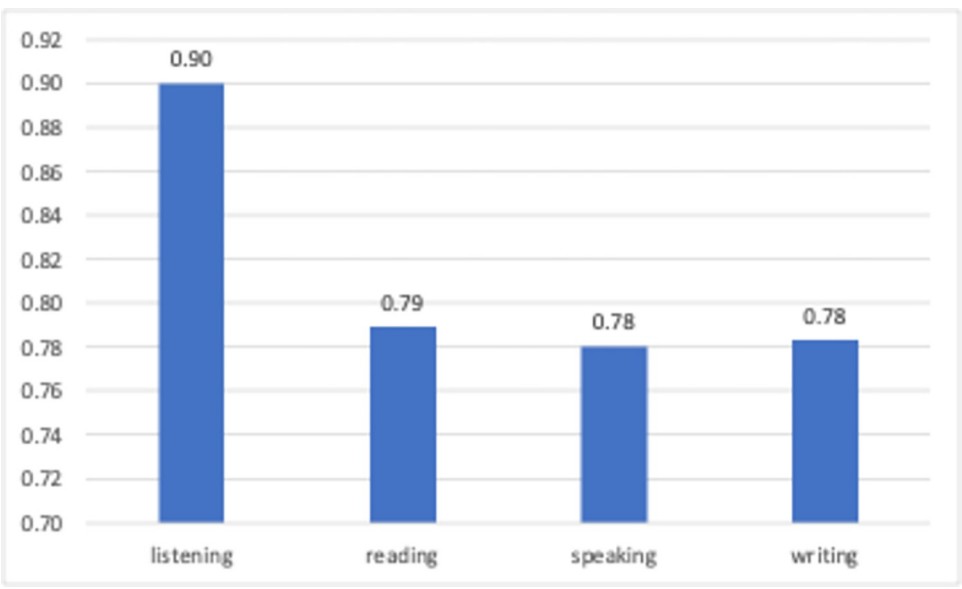

**Fig 12. Average sentiment scores of language learning categories.**

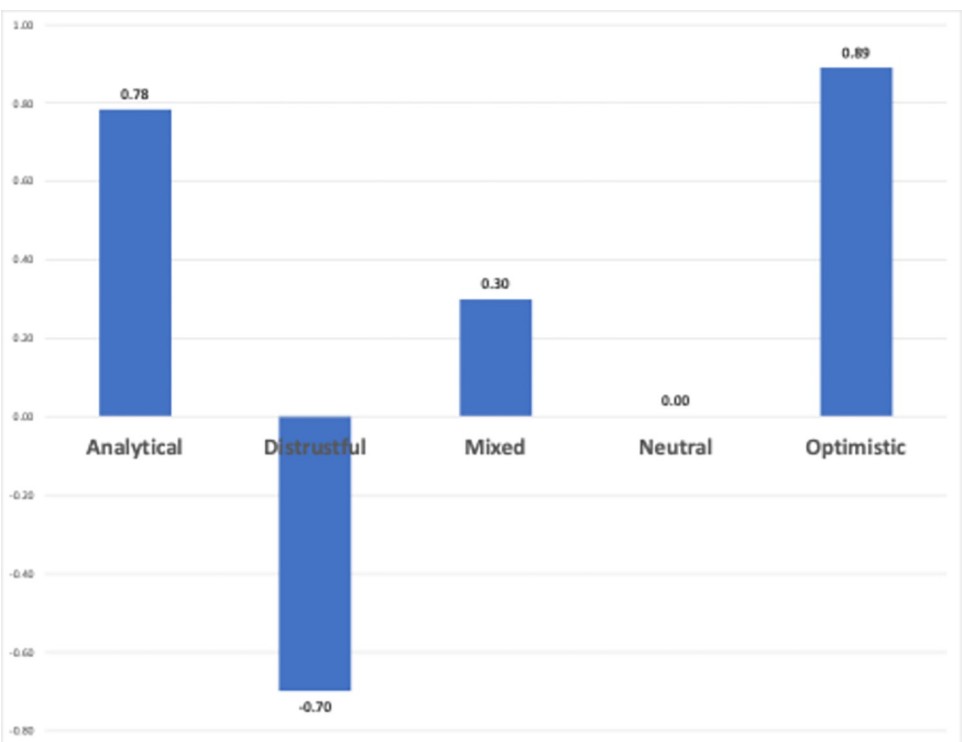

**Fig 13. Average sentiment scores vs sentiment classes.**

Fig 14 displays the average sentiment scores for clusters that focus on different language skills. The cluster "Basic expression skills" has an average sentiment score of 0.77, indicating a generally positive sentiment. The cluster "Intercultural communication skills" has an average sentiment score of 0.50, indicating a moderate positivity. The analysis indicates that videos on intercultural communication have a more sentimentally balanced impression. Additionally,

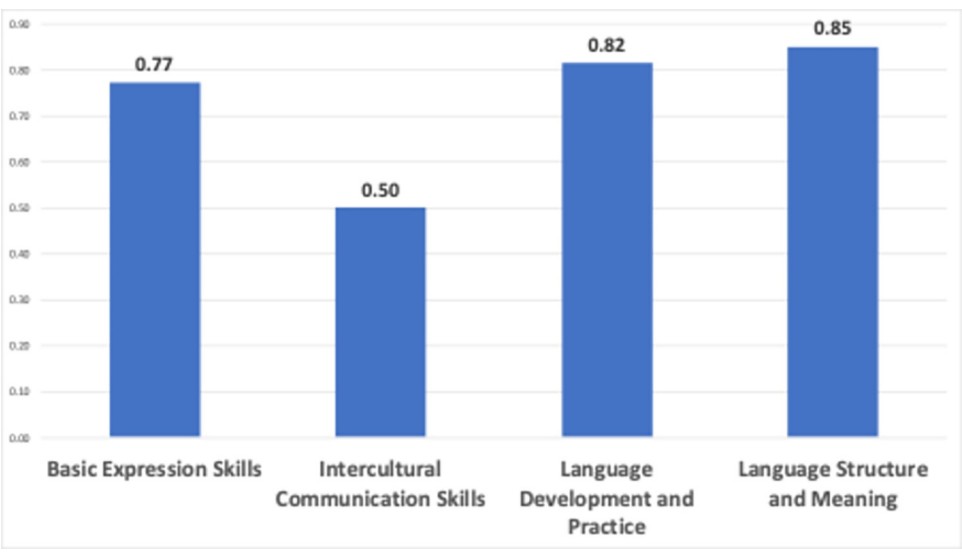

**Fig 14. Average sentiment scores vs clusters.**

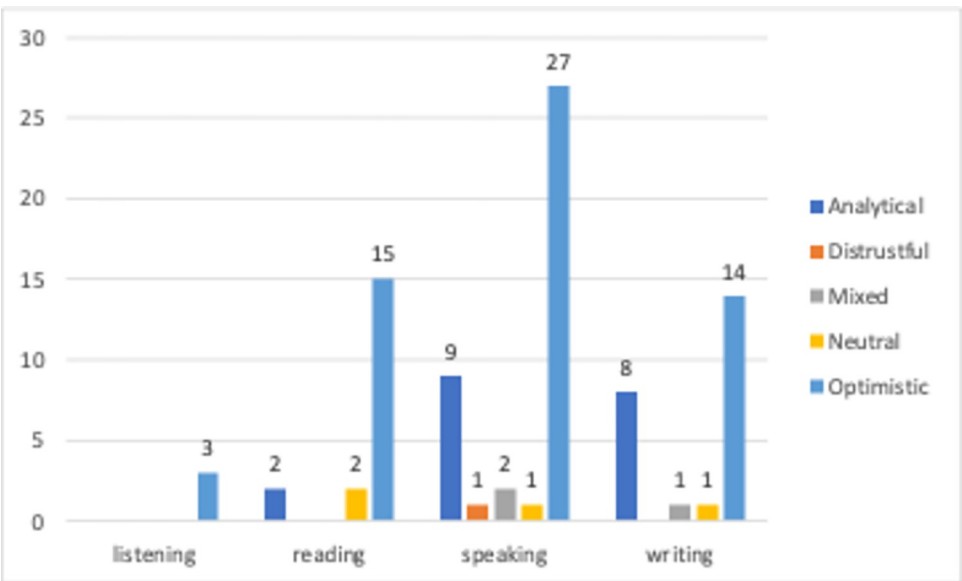

**Fig 15. Language learning categories vs sentiment classes.**

the "Language development and practice" cluster has a more positive impact than the basic expression skills and intercultural communication skills clusters. The "Language structure and meaning" cluster has the highest value, indicating that videos focusing on language structure and meaning have a very positive impact on video creators.

Fig 15 shows the distribution of basic language skills by sentiment classes. The sentiments of the videos for different skill areas are clearly different. The three videos associated with the listening skill are linked only to the optimistic emotion, indicating that content providing listening practice is usually presented in a positive atmosphere. Of the 19 videos associated with the reading skill, two are linked to analytical, two to neutral, and 15 to optimistic emotions. The text suggests that reading is typically portrayed positively, with the aim of motivating and providing an enjoyable learning experience. Out of the 40 videos related to speaking skills, only 5 were associated with different emotions. The videos that include speaking practice predominantly have an optimistic tone, indicating that content aimed at improving speaking skills is usually presented positively. Of the 24 videos related to writing skills, 8 are associated with analytical, 1 with mixed, 1 with neutral, and 14 with an optimistic tone. The prevalence of an optimistic tone in the videos involving writing practice suggests that they aim to convey to the viewers that improving writing skills is an enjoyable and positive process.

Fig 16 shows the analysis of sentiments associated with four different clusters defined by experts in 66 videos on the use of generative AIs in language learning. Videos that focus on basic expression skills exhibit high levels of optimism, indicating that the content in this area is positive and motivating. The intercultural communication skills cluster's emphasis on objectivity suggests that information-oriented content may become more prominent in intercultural communication. It is important to note that there is no optimistic sentiment in this cluster. Upon examining the videos in this cluster, it was found that four out of the six videos were uploaded by the same person, Toğrul Əzizov. (https://www.youtube.com/@togrul_ezizov). All videos uploaded by him reflect an analytical stance towards generative AIs in language learning, and his shot videos on how ChatGPT could be used for language learning and he showed how to give prompts to Chat GPT and demonstrates various examples. Reaming

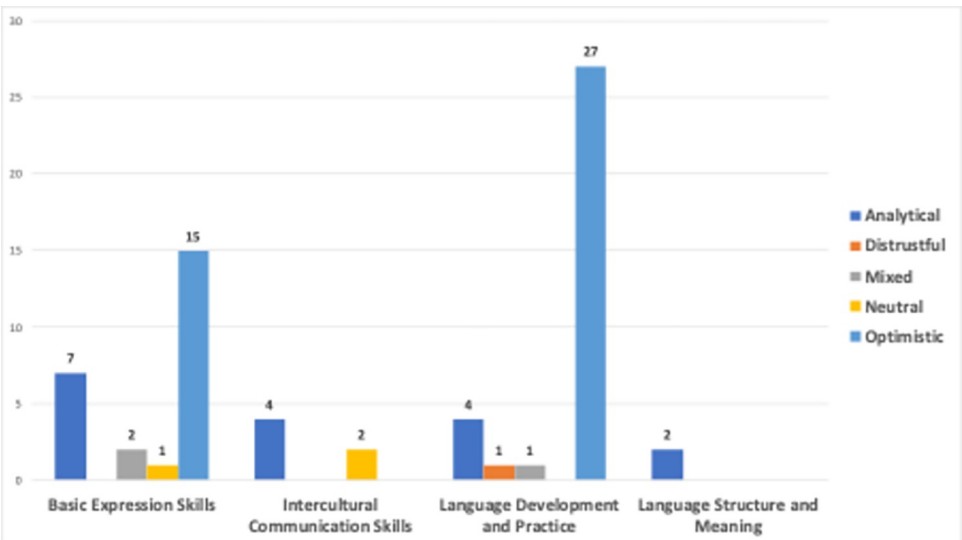

**Fig 16. Sentiment classes vs clusters defined by experts.**

two videos in this cluster have neutral sentiment. The language development and practice cluster appear to have a dominantly optimistic sentiment, suggesting that the videos in this cluster aim to motivate viewers positively. However, unlike the other clusters, one video related to this cluster, titled "ChatGPT just killed language learning", presents a distrustful tone as explained earlier.

Based on the analysis presented in Fig 17, it has been determined that videos focusing on the use of generative AIs in language learning exhibit a heterogeneous distribution among different language skills and the expert clusters identified for these skills. Videos associated with

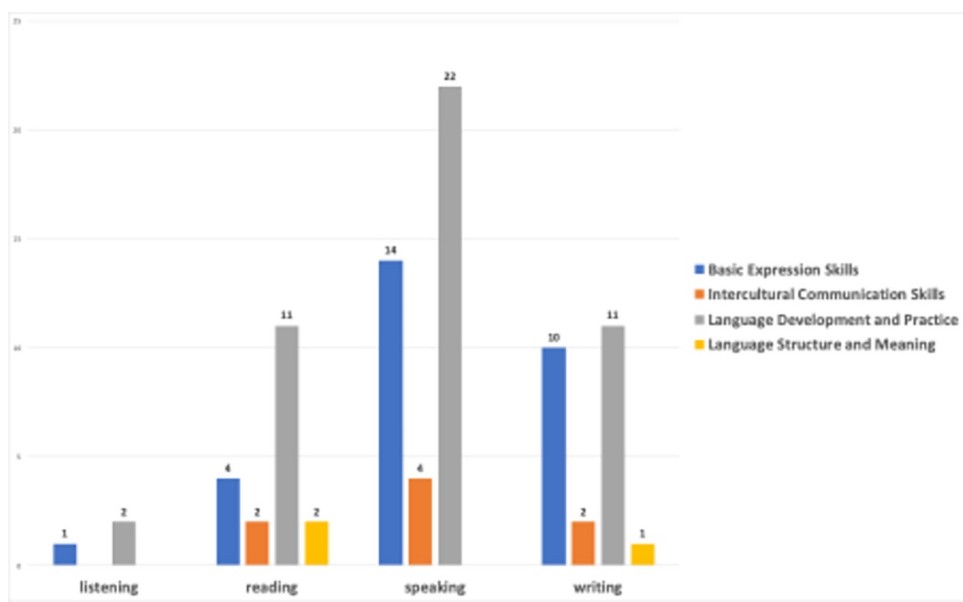

**Fig 17. Language learning categories vs expert clusters.**

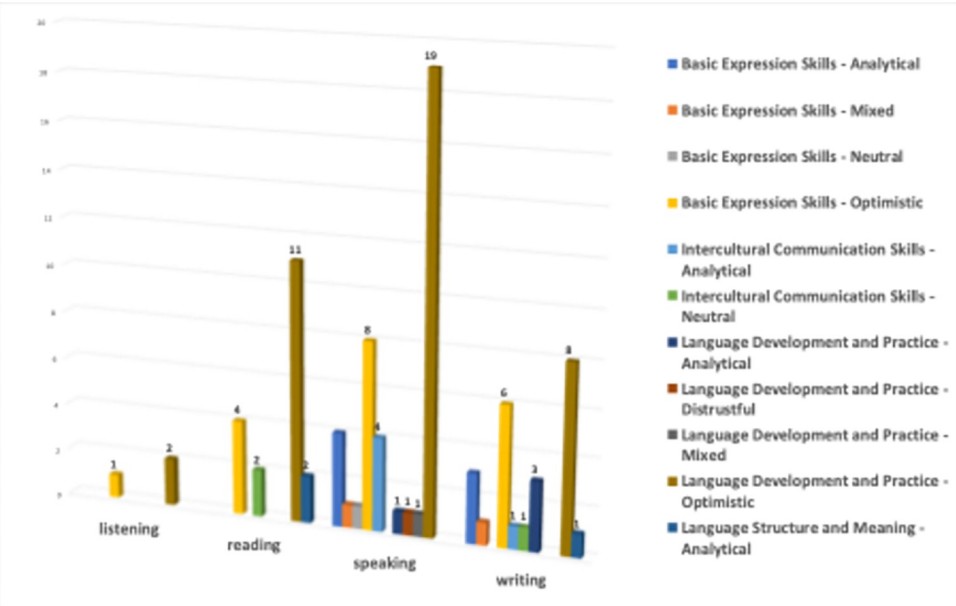

**Fig 18. Category vs cluster vs classes.**

listening skills belong to the clusters of basic expression skills and language development and practice. The videos associated with basic language expression skills and language practice are only linked to these two clusters. In contrast, videos related to reading skills are generally associated with four clusters. This may be because they aim to provide viewers with a more diverse language learning experience. The videos related to speaking skills outnumber those related to other skills, and are focused on three associated clusters. Similarly, videos related to writing skills are associated with four clusters, which is comparable to reading skills.

Finally, the multiple comparison of categories, clusters, and classes are depicted in Fig 18.

Most videos on listening skills have an optimistic tone. This indicates that content that focuses on listening practice and language development generally reflects positive sentiments. These videos, which are related to "basic expression skills" and "language development and practice" as a cluster, may aim to strengthen basic expression skills and provide positive views in the language learning process. In videos related to reading skills, various clusters evoke different sentiment. Four videos in the basic expression skills cluster seem to aim at strengthening fundamental language expression skills and creating a positive atmosphere by reflecting optimism. Meanwhile, two videos in the intercultural communication skills cluster suggest content focused on intercultural communication with a neutral tone. The 11 videos in the "language development and practice" cluster generally have optimistic sentiments, presenting a positive approach to language development. The two videos in the "language structure and meaning cluster", on the other hand, may be associated with a more analytical approach, aiming to emphasize the structure and meaning of language.

Videos associated with speaking skills, linked to the "basic expression skills" and "language development and practice" clusters, reflect not only optimistic sentiments but also mixed, neutral, analytical, and distrustful sentiments as well. This variety indicates that videos focusing on speaking skills and associated with these clusters are not only characterized by a positive atmosphere but also by critical thinking and complex emotions. Furthermore, it is observed that content emphasizing intercultural communication skills is generally approached with a more analytical perspective. It can be said that content aimed at providing in-depth thinking and

analysis opportunities in intercultural interaction is included in this cluster. Videos associated with writing skills are identified as having an analytical sentiment across all clusters. Therefore, it can be said that videos aimed at writing skills encompass content approached with a critical and analytical perspective across all clusters. One video is associated with a mixed sentiment in the basic expression skills cluster, while another is linked to the intercultural communication skills cluster with a neutral sentiment. This indicates that videos aimed at writing skills focus on different topics associated with these two clusters.

## 4. Discussion

The purpose of this study is to conduct a detailed examination of videos published on YouTube from December 2022 to October 2023 that discuss the use of generative AI in language learning. The study examines the findings collected under four primary headings: video content and basic language skill category, video content and opinion/sentiment classification, video content and clusters, and cross-sectional analysis. A detailed examination of the findings under these headings aims to contribute to an understanding of the content, interaction and trends in YouTube videos on the use of generative AI in language learning. The literature contains studies on the application of generative AI in language learning [68], assessment and evaluation [69], content generation for reading skill development [70], and writing assistance [71]. This study differs from others in the literature by covering all aspects related to the use of generative artificial intelligence for basic language skills on YouTube, which is considered an informal learning platform. Additionally, sentiment analysis was employed to capture the stance of the analyzed YouTube videos towards basic language skills. Previous studies have focused primarily on vocabulary rather than language learning. Furthermore, this study distinguishes itself from other literature by analyzing YouTube videos. This study contributes to the literature by comprehensively analyzing YouTube videos to examine the use of generative AI in language learning. By incorporating both topic modelling and sentiment analysis, it provides insights into the attitudes towards language learning in these videos. However, the study is limited to YouTube videos and does not include content from other online platforms. Additionally, the analyzed videos are restricted to a specific time period.

This study aimed to answer how YouTube videos about generative AI in language learning specifically address different language skills such as speaking, listening, reading, and writing. The analysis of the distribution of video content by category indicates that most videos are related to speaking skills. This suggests that examined videos prioritize the development of language skills for everyday use and emphasize the speaking process. The literature emphasizes the development of oral communication skills, particularly speaking, as the most desirable skill for students [72,73]. Writing skills are also highly valued and rank second. These skills encompass both written and verbal forms of communication. However, there are fewer videos that focus on listening and reading skills, indicating that language learning video content places less emphasis on receptive language processes than on productive language skills. However, [72] found that speaking is the most important skill for language learners, followed by listening, writing, and reading. This highlights the need for balanced language teaching materials that provide students with a comprehensive language experience. The prominent words for each category also reveal important insights. The videos on how generative AI could be used for language learning covered the basic language skills from a broad perspective. Therefore, content that provides learners with interactive learning experiences and aims to improve their language skills is crucial for developing strong writing abilities. The importance of vocabulary in effective writing is well-established in the literature [74–78]. This relationship between reading, writing, and vocabulary has been supported by [77]. According to research, writing skills

can be improved through the acquisition of a rich vocabulary, which in turn positively impacts the development of reading skills. It reflects a mastery of appropriate language use in different speech situations and contributes to fluent speech [79–81]. In speaking, a rich vocabulary is essential for effective communication and expressing ideas or feelings. The vocabulary used to develop reading skills aims to assist students in comprehending word meanings, interpreting texts, and understanding multilingual content. The literature has discussed the relationship between reading skills and vocabulary in terms of text comprehension [82,83]. Therefore, this study's finding on reading skills aligns with the existing literature. This finding meets the expected relationship between reading skill and vocabulary for text comprehension. The words related to listening skills indicate that content necessary for developing other language skills can be presented through listening. This argument is partially supported by [84]. According to research, there is not as strong a correlation between listening skills and vocabulary as there is between reading and writing skills. However, vocabulary is still necessary for developing listening skills. The findings indicate that YouTube videos on generative AI prioritize productive language skills, particularly speaking and writing, over receptive skills like listening and reading. This imbalance highlights a critical need for more comprehensive language learning resources that address all language skills. However, a limitation of this study is that it primarily focuses on English-language videos, potentially overlooking insights from non-English content.

The second research question guiding this study was to determine the distribution of sentiments—optimistic, distrustful, mixed, analytical, ethical, biased, futuristic, and neutral—expressed in YouTube videos about the use of generative AI for language learning. The sentiment analysis of videos posted on YouTube about the use of generative AI in language learning shows mostly positive results. Other studies have also found that language learning through YouTube can be effective [85–89]. However, [90] cautioned that YouTube content should not be relied upon completely in language learning as it cannot always be controlled. Most videos with an optimistic tone show that generative AI for language learning has a positive stance. Furthermore, videos with an analytical tone provide detailed information about the use of generative AI. Videos with mixed or neutral sentiments demonstrate the variety of attitudes towards the use of generative AI. On the other hand, the low number of videos associated with a distrustful sentiment indicates that there is no dominant distrust among viewers regarding generative AI. Therefore, it might be concluded that generative AI in language learning videos generally leave a positive impression and give confidence in content creators. The usefulness of generative AI for language learning is emphasized by [56]. However, their analysis of YouTube videos on the impact of ChatGPT on language learning revealed a negative finding. Many YouTubers concluded that ChatGPT is not reliable enough in explaining grammar and rules. Additionally, there is diversity in sentiment values. Sentiment values regarding the use of generative AI in language learning vary, but the general trend is positive. The study conducted by [57] also found positive results in their study on language development with generative AI on YouTube, supporting the findings of this study. However, they noted that for generative AI to be effective, users must be technically competent. The analysis of the comment counts indicates low interaction with language learning and generative AI in language learning videos. Most videos received no comments, indicating a lack of feedback and interaction from viewers. Only a small number of videos attracted attention and interaction from a specific audience. The sentiment analysis of YouTube videos on generative AI in language learning shows mostly positive results, consistent with other studies on YouTube's effectiveness. Optimistic videos highlight a positive stance, while analytical ones provide detailed insights. The overall positive trend suggests broad acceptance.

Thirdly, this research aimed to discover how YouTube video content is clustered based on wording using text mining methodologies. The analysis results indicate that videos on

language learning and generative AI are categorized into four distinct clusters. The cluster with the highest number of videos, "language development and practice", aims to assist viewers in expanding their vocabulary, learning grammar rules, and using language in real-life situations. This finding is consistent with the literature [52,91,92]. The cluster named "basic expression skills" concentrates on teaching forms of expression, methods, and techniques. On the other hand, the "intercultural communication skills" cluster provides guidance to language learners on how to interact with different cultures and use their language skills in this context. Lastly, the "language structure and meaning" cluster focuses on the structure, meaning, and in-depth analysis of language. It is worth noting that this cluster has the lowest number of videos. It might indicate that content creators' interest and audience demand are concentrated on specific language skills or subject areas. As topics become more complex, the number of videos decreases, which contradicts the Common European Framework of Reference (CEFR) [93]. According to the CEFR, higher language levels (C1 and C2) require greater competence in the in-depth understanding and use of language. At these levels, language users are expected to understand complex texts and possess advanced communication skills. They are also expected to have a deeper understanding of how to use language in cultural and social contexts. The analysis reveals that YouTube videos on language learning and generative AI are categorized into four distinct clusters, with "language development and practice" being the most prevalent. This consistency with existing literature underscores the focus on practical language skills. However, the limited number of videos in the "language structure and meaning" cluster suggests a gap in content addressing advanced linguistic analysis, which is critical according to the CEFR for higher language proficiency levels. Additionally, the categorization into clusters is based on the content available at the time of the study, which may not reflect all emerging trends and new content that could provide further insights into language learning and generative AI.

Finally, this study aimed to present the results of cross-sectional evaluations of the clustering, classification, and sentiment analyses of YouTube video contents about generative AI in language learning. The findings on monthly distribution of generative AI language learning videos show diverse trends in basic language skills. Limited content on listening skills compared to others underscores room for improvement in AI applications. While videos on other skills appear across most months, their distribution varies rather than steadily increasing. This finding is consistent with the literature on listening skills, which have been relatively neglected compared to other skills [94–97]. The regular provision of content for speaking skills demonstrates the overall popularity and engagement of generative AI-powered applications in this domain. The delivery of consistent content in reading may indicate that generative AI in text-based learning tends to perform more strongly and consistently. In their study, [98] found no significant difference in reading performance between students with and without the AI facilities. Fluctuations in writing skills may indicate a changing trend depending on factors such as content variety, learner feedback, or practice opportunities. Additionally, writing is a complex task that requires various skills [99]. In the findings for sentiment analysis, the videos mostly reflect positive and optimistic sentiments. In the first month, only the analytic theme attracted attention. In the following months, it is seen that efforts to diversify the audience intensified, especially during periods when different emotion themes coexisted. At the same time, the video with the theme "distrustful" in September 2023, which reflects a critical point of view, can be interpreted as an effort to present different perspectives to the audience. However, there is a slight fluctuation in the sentiment polarities. In addition, the small number of contents may have reduced the likelihood of negative sentiments coming to the fore. The higher number of videos for other skills indicates that a wider range of content is presented in these areas. According to the [93], it aims to improve students' language skills through its

progressively presented language levels and topics suitable for each level. Since each of these stages is important for the next stage, students are expected to approach each level with the same care. In terms of CEFR levels, content creators' sentiments towards the clusters in this study could be considered as a desired and expected situation. Based on the findings regarding the distribution of language learning categories and clusters, there is not a balanced distribution among language skills, and significant differences exist among different clusters. Particularly, a distinct situation has been identified in the listening and speaking skills, which are two elements of verbal communication. Videos related to listening skills create a positive emotional impact by focusing on basic expression skills and language practice. However, there is no relationship between speaking skills and the language structure and meaning cluster. The study found that sentiments vary among clusters that focus on different language skills. Specifically, the "Basic Expression Skills" cluster showed that videos emphasizing basic expression skills with a high average sentiment value generally have a positive sentiment polarity. This suggests that content targeting this cluster aims to support learning by providing viewers with a positive experience. The cluster of "Intercultural Communication Skills", however, leaves a more balanced sentimental impression with a moderate level of positivity. This may be due to the fact that the content on intercultural communication relies more on information sharing. In contrast, the "Language Development and Practice" cluster has a more positive sentiment polarity compared to the "Basic Expression Skills" and "Intercultural Communication Skills" clusters. The positive situation could be attributed to the fact that it focuses on practical knowledge necessary for language teaching. "The Language Structure and Meaning" cluster, which has the highest average sentiment value, creates a significantly positive emotional impact on viewers with videos focusing on the structure and meaning of language. Content that focuses on the structural features and meaning of language aims to provide viewers with a more analytical thinking process. Positive sentiments generally prevail in content related to language skills. Learners' attribute equal importance to topics associated with these clusters. On the other hand, the fact that videos related to reading skills are associated with four different clusters indicates an intention to support language development in a multifaceted manner by offering viewers a wide range of topics. Writing skills, like reading skills, are associated with different clusters. Finally, based on the findings related to category, cluster, and sentiment analysis, the fact that videos related to listening skills are generally associated with optimistic emotions indicates that such content is presented in a positive atmosphere focusing on listening practice and language development. It is believed that videos associated with the "Basic Expression Skills" and "Language Development and Practice" clusters aim to strengthen basic expression skills and provide positive language learning experiences. In videos associated with reading skills, the presence of different clusters and emotions is remarkable. It is observed that videos in the "Basic Expression Skills" cluster aim to strengthen the basic expression skills of the language and evoke positive emotions. Videos in the "Intercultural Communication Skills" cluster, on the other hand, contain a neutral emotion, drawing attention to content focused on intercultural communication. Videos in the "Language Development and Practice" cluster generally contain optimistic sentiments. It can be said that videos in the "Language Structure and Meaning" cluster aim to emphasize the structure and meaning of the language. Videos associated with speaking skills contain not only optimistic sentiments but also critical thinking and complex emotions. Videos focusing on the "Intercultural Communication Skills" cluster are generally aimed at in-depth analysis of intercultural interactions with an analytical perspective. In videos associated with writing skills, analytical sentiment is prominent in all clusters, indicating that writing skills are approached with a critical and analytical perspective. Additionally, writing skills are associated with the "Basic Expression Skills" and "Intercultural Communication Skills" clusters, focusing on various topics. The analysis of generative AI language learning

videos on YouTube reveals diverse trends across basic language skills, with a significant focus on speaking and reading skills, indicating the popularity and engagement in these areas. However, the limited content on listening skills highlights a gap, suggesting a need for more resources in this domain. The sentiment analysis shows a predominantly positive outlook, especially in clusters related to basic expression and language development, though writing skills reflect a more critical perspective. A limitation of this study is that it relies on publicly available YouTube videos, which may not comprehensively capture all viewpoints and trends in language learning with generative AI.

## 5. Conclusion

The aim of this study was to investigate how generative AI and language learning were addressed in YouTube video content, specifically examining how language skills are approached in the context of using generative AI. Additionally, the study analyzed the sentiments of YouTube content creators regarding the potential use of generative AI for language learning. The study's findings might guide the development of AI-supported materials in the field of language teaching. The majority of the videos related to the use of generative AI in language teaching focus on speaking and writing skills. Language learning videos aim to develop language skills for daily use, with an emphasis on speaking. However, there are fewer videos that focus on listening and reading skills, indicating less emphasis on receptive language processes. The analysis of AI-supported language teaching videos reveals that they cover basic language skills from a broad perspective. The sentiment analysis results indicate that these videos generally have a positive outlook, with prominent expressions of optimism and analytical thinking. According to the results of cluster analysis, videos have been divided into four different clusters: language development and practice, basic expression skills, intercultural communication skills, and language structure and meaning. It is observed that videos belonging to each cluster provide students with the opportunity to develop different language skills and understanding. The results of sentiment analysis indicate that the videos generally leave a positive impression.

## 6. Recommendations

Based on the study's findings, it is important to broaden the video content to encompass additional language skills, including listening and reading, in addition to speaking and writing. This could offer learners a more comprehensive language learning experience. To tackle the problem of low engagement, content creators should consider various methods of engaging with viewers, such as question and answer sessions, interactive exercises which might increase engagement. It is recommended that language learners explore a variety of platforms beyond YouTube to gain exposure to new tools and technologies. Additionally, offering supplementary content related to critical thinking, analytical skills, or complex emotions could further enrich the learning experience.

According to the cluster analysis, it is important for the videos to emphasize subject areas that focus on strengthening specific language skills. Content producers can enhance the learning experience by clearly communicating to viewers which language skills they aim to develop. Encouraging audience participation is crucial to receive more comments and feedback on videos. Adding interactive elements, such as polls or prompts for viewer feedback, can encourage engagement and participation. This study aimed to conduct sentiment and opinion analysis of selected YouTube videos on the use of generative AI tools for language learning. Further studies could extend this research by conducting content analysis of YouTube videos to provide more insight and capture trends.

## 7. Limitations

The study's dataset comprises 66 videos obtained through the YouTube API and filtered according to inclusion criteria aligned with the study's objectives. The study is restricted to English-language videos available until October 2023 that met the inclusion criteria. Therefore, it may not fully reflect linguistic diversity. This study focuses on videos covering Generative AI, Chat GPT, and language learning. It is important to note that topics or approaches outside of this scope may have been omitted. Videos with very short durations (0–15 seconds) were excluded, which may make it difficult to evaluate content with a certain amount of continuity and may prevent a more comprehensive analysis. The study solely analyzed video comments numerically, potentially overlooking factors that influence the content and diversity of these comments.

## Supporting information

**S1 Dataset. Data obtained from YouTube videos.** The data obtained from YouTube videos for the purpose of this study are openly available at https://osf.io/qy328/?view_only=a3b1b 509673d4d72988f40bbae82a8e4.
(XLSX)

**S1 Appendix.**
(DOCX)

## Author Contributions

**Conceptualization:** Mazhar Bal, Ayşe Gül Kara Aydemir, Mustafa Coşkun.

**Data curation:** Mazhar Bal, Ayşe Gül Kara Aydemir, Mustafa Coşkun.

**Formal analysis:** Mazhar Bal, Ayşe Gül Kara Aydemir, Mustafa Coşkun.

**Methodology:** Mazhar Bal, Ayşe Gül Kara Aydemir, Mustafa Coşkun.

**Project administration:** Mazhar Bal, Ayşe Gül Kara Aydemir, Mustafa Coşkun.

**Resources:** Ayşe Gül Kara Aydemir.

**Software:** Mustafa Coşkun.

**Supervision:** Mazhar Bal, Ayşe Gül Kara Aydemir, Mustafa Coşkun.

**Validation:** Mazhar Bal, Ayşe Gül Kara Aydemir, Mustafa Coşkun.

**Visualization:** Mazhar Bal, Ayşe Gül Kara Aydemir, Mustafa Coşkun.

**Writing – original draft:** Mazhar Bal, Ayşe Gül Kara Aydemir.

**Writing – review & editing:** Mazhar Bal, Ayşe Gül Kara Aydemir, Mustafa Coşkun.

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
