## [Decision Letter · Decision Letter 0]

30 May 2024

PONE-D-24-17175Exploring YouTube content creators' perspectives on generative AI in language learning: Insights through opinion mining and sentiment analysis

PLOS ONE

Dear Dr. Kara Aydemir,

Thank you for submitting your manuscript to PLOS ONE. After careful consideration, we feel that it has merit but does not fully meet PLOS ONE’s publication criteria as it currently stands. Therefore, we invite you to submit a revised version of the manuscript that addresses the points raised during the review process.

We look forward to receiving your revised manuscript.

Kind regards,

Dilrukshi Gamage, Ph.D

Academic Editor

PLOS ONE

2. We note that Figure 4 in your submission contain copyrighted images. All PLOS content is published under the Creative Commons Attribution License (CC BY 4.0), which means that the manuscript, images, and Supporting Information files will be freely available online, and any third party is permitted to access, download, copy, distribute, and use these materials in any way, even commercially, with proper attribution. For more information, see our copyright guidelines: http://journals.plos.org/plosone/s/licenses-and-copyright.

1. You may seek permission from the original copyright holder of Figure(s) [#] to publish the content specifically under the CC BY 4.0 license.

Additional Editor Comments:

Thank you for submitting this insightful paper. While the manuscript, "Exploring YouTube content creators' perspectives on generative AI in language learning: Insights through opinion mining and sentiment analysis," provides a valuable analysis of YouTube videos on generative AI in language learning, it requires significant revisions before it can be considered for publication despite the reviewers has recommend for Minor Revisions. Reviewers may have missed major details and seems need more rigorous review.

Firstly, the introduction, while comprehensive, needs a broader discussion on various generative AI models beyond ChatGPT and a clear justification for the exclusive focus on ChatGPT. The research questions, especially RQ1, lack clarity and require rephrasing to make the expectations more explicit.

The methods section, though detailed, leaves several critical points ambiguous. The role of video comments in the analysis is not clear, and there is confusion about whether these comments were used to reflect the perspectives of content creators or general users. Additionally, the classification and sentiment analysis processes need more transparency. There is no mention of accuracy checks for the classification models used, and the absence of code in the supplementary materials undermines the reproducibility of the study.

The results section does not adequately relate the research questions to the outcomes. Figures 6 and 7, for example, are confusing due to the unclear units of analysis and the lack of detail about what is being measured. Moreover, the sentiment analysis lacks transparency regarding the models and libraries used, such as whether VADER or NLTK was employed. The manuscript should provide examples and include the code in the appendix to enhance transparency.

The discussion section is currently disorganized and fails to effectively highlight the impact of the study. It needs to be restructured to address each research question separately and clearly. The discussion should also include recommendations and limitations, which are currently missing.

To improve the paper, the authors should:

Broaden the introduction to include various generative AI models and justify the focus on ChatGPT.

Clarify and rephrase RQ1 to make the expectations explicit.

Clearly explain the role of video comments and whether they reflect content creators' or general users' perspectives.

Provide transparency in the classification and sentiment analysis processes, including accuracy checks and the inclusion of code.

Relate the results directly to the research questions and clarify the units of analysis in the figures.

Organize the discussion to address each research question separately, include recommendations and limitations, and end with a strong conclusion.

Given the extent of these necessary revisions, I recommend that this paper undergo a major revision cycle.

Reviewers' comments:

Reviewer's Responses to Questions

**Comments to the Author**

1. Is the manuscript technically sound, and do the data support the conclusions?

Reviewer #1: Yes

Reviewer #2: Yes

Reviewer #3: Partly

2. Has the statistical analysis been performed appropriately and rigorously? 

Reviewer #1: No

Reviewer #2: Yes

Reviewer #3: No

3. Have the authors made all data underlying the findings in their manuscript fully available?

Reviewer #1: Yes

Reviewer #2: Yes

Reviewer #3: No

4. Is the manuscript presented in an intelligible fashion and written in standard English?

Reviewer #1: Yes

Reviewer #2: Yes

Reviewer #3: Yes

5. Review Comments to the Author

Reviewer #1: This paper delves into the perspectives of YouTube content creators regarding the utilization of generative AI in language learning. Its primary merit is the thorough exploration of these creators' views on the topic, achieved through innovative analysis that emphasizes practical approaches to language learning. Nevertheless, the discussion on how generative AI has enhanced language learning could have been more detailed and explicit

Reviewer #2: Strengths:

-The introduction provides a thorough background on the importance of technology in language learning, highlighting the role of generative AI.

-It effectively contextualizes the study within the existing literature, mentioning various technologies previously used in language learning.

-The rationale for focusing on YouTube content creators is well-explained, emphasizing the platform's reach and influence.

Some suggestions to improve the manuscript can be

-The introduction could benefit from a more explicit statement of the research gap. While the context is well-established, the specific gap this study addresses could be clearer.

-The research questions could be presented more prominently to outline the study's objectives clearly.

-The manual revision of model results by researchers introduces potential bias (line 197-198). Discussing measures taken to minimize this bias or an inter-rater reliability metric would strengthen the methodology.

Reviewer #3: Thank you for the insightful paper. I really enjoyed reading it. Overall, the paper is well-written, easy to follow, and the key objectives, methods, and discussions are clearly presented and easy to understand. The perspectives on GenAI are trending in the research domain, and the authors of this paper particularly examine YouTubers' perspectives on language learning by analyzing their video content.

In doing so, the authors specifically used the ChatGPT model while incorporating NLP methods. Although there is a growing body of research and interest, and the authors have highlighted valuable insights, let me point out some areas that need to be addressed.

The introduction is clear and effectively sets the stage for the research. However, some parts of the introduction are dense and difficult to follow. It would have been beneficial to specify a broader range of generative AI models beyond ChatGPT and justify why only ChatGPT was chosen for this study. Although the research questions are clear, it would be more effective if you could elaborate on the expectations of RQ1. Initially, it was difficult to grasp, and it seems to need some rephrasing:

“RQ1: How are YouTube video contents about the use of generative AI in language learning related to language skills?”

When you ask about the relationship with language skills, what were the expectations? This is not clear in the introduction.

The methods are clear, and it seems a rigorous process was followed to filter the specific videos. However, I am confused about whether you used the comments of the videos for analysis. The statement "After accessing and filtering videos, video comments were collected using a relevant API" suggests that comments were used. Since comments are from general users and not from creators, they may not provide the perspectives outlined in the paper.

On the other hand, I assume you are using classification to answer RQ1, but learning skills related to reading, writing, listening, and speaking components were never introduced earlier to clarify if that was the expectation of the relationship. Additionally, when you used the model, did you check the accuracy levels? There was no indication of your code in the OSF, but the Excel sheet already categorizes the data. To increase confidence in your findings, please provide the code you used for the task.

You performed a topic model using LDA. Was the purpose of this to understand the topical areas of the YouTube transcripts, or were comments included as well? What is the transparency of the sentiment model? What library was used—VADER or NLTK? Check the code in the prompt and be open about everything. Specifically, if you can analyze the sentiment into categories such as Optimistic, Distrustful, Mixed, Analytical, Ethical, Biased, Futuristic, Neutral, provide evidence of this robustness. Show examples and include the code in the appendix.

The filtration process could have been explained a bit more to give a clearer idea. Which API was used to collect video comments? What was the reason for choosing the defined prompts? Were any previous evaluations done on that?

In the results section, it would have been helpful to relate the research questions to the outcomes.

There are a few issues to address. When you inserted Figures 6 and 7, the unit of analysis for the sentiment is confusing. Did you analyze sentiment at the YouTube video level? What is the y-axis representing? At some points, you mentioned comment sentiment, but it was not clear in the methods how you would be using the content—whether at the transcript sentence level, paragraph level, or per video transcript level. If you used comments, what was the unit of analysis, per video level?

For Figure 12, which shows average sentiment scores, what specifically is being averaged? Did you take all the transcripts related to listening and average them by the number of what? Please be more specific in your descriptions. Additionally, none of these details were explicitly mentioned in the methods.

The discussion section needs to be organized to highlight each research question in subparagraphs. Currently, everything is in one plane, making it difficult to distinguish the impact of the study. Add the recommendations and limitations under the Discussion section and finish with the conclusion.

I believe if you can address the mentioned issues in the paper, it has the potential for the next round of review.

6. PLOS authors have the option to publish the peer review history of their article (what does this mean?). If published, this will include your full peer review and any attached files.

Reviewer #1: No

Reviewer #2: No

Reviewer #3: **Yes: **Naveen Periyasamy Rajan

---

## [Author Response · Author response to Decision Letter 0]

4 Jul 2024

Dear Dr. Dilrukshi Gamage,

I am writing to submit the revised version of our manuscript titled “Exploring YouTube content creators' perspectives on generative AI in language learning: Insights through opinion mining and sentiment analysis” by Mazhar Bal, Ayşe Gül Kara Aydemir, Mustafa Coşkun for reconsideration for publication in PLOS ONE. We have carefully considered the editor’s and reviewers’ comments and suggestions. We deeply appreciate the time and effort dedicated by the reviewers and editorial board in evaluating our manuscript. We believe that the revised manuscript now better meets the standards of PLOS ONE and addresses all the reviewers' concerns comprehensively.

Based on the comments, we have made several key revisions. The introduction section has been expanded to include a broader discussion of various generative AI models beyond ChatGPT. This expansion includes a clear justification for focusing on ChatGPT due to its widespread adoption and rapid rise in popularity. Additionally, the literature on generative AI in language learning has been extended. We clearly identified the research gap and expanded both the literature review and discussion within the introduction section. Our research questions have been revised for better clarity. In the methods section, we addressed a misunderstanding caused by a mistake in the model prompt, where "comment" was incorrectly used instead of "transcription," and we have corrected this. We justified using GPT-4-0613 for our analysis due to its high accuracy level and the extensive data it is trained on. The execution progress can be seen in the attached code of sentiment analysis in Appendix A. The results section was revised to connect each research question with the findings explicitly, and detailed explanations for Figures 6 and 7 were provided. The discussion section was reorganized to address each research question separately, with recommendations and limitations included.

Regarding Figure 4, we would like to clarify that the images in this figure are not sourced from any third-party stock image collection. Instead, these images were generated by us using the wordcloud module in Python, based on our own text dataset. As the creators of these images, we hold the exclusive rights to them. Since the images were not copied from any external source, there are no additional copyright considerations or permissions required beyond our own authorization as the original creators. We hereby grant PLOS the right to use, distribute, and publish these images under the Creative Commons Attribution License (CC BY 4.0).

Additionally, in response to the data availability comments, we confirm that our Supporting Information files do not contain any identifying data. Our study utilized transcripts of publicly available videos on YouTube, which are already accessible to the general public and do not contain any identifying information beyond what is already publicly shared by the content creators themselves. We have ensured that no indirect identifiers that could potentially compromise participant privacy are included in our data. Our data sharing practices are in full compliance with YouTube Terms of Services, YouTube Developer Policy, YouTube API Policy, Google Terms of Service, and relevant legislation. 

I appreciate your time and reconsideration and look forward to the opportunity of having our work reconsidered for publication in PLOS ONE. We look forward to your favorable consideration of our revised submission. Ayşe Gül Kara Aydemir is the corresponding author of this study. Should you have any questions or require further information, please contact me via email at aysegulkara@akdeniz.edu.tr.

Sincerely,

Ayşe Gül Kara Aydemir

---

## [Editor Report · Decision Letter 1]

17 Jul 2024

Exploring YouTube content creators' perspectives on generative AI in language learning: Insights through opinion mining and sentiment analysis

PONE-D-24-17175R1

Dear Dr. Kara Aydemir,

We’re pleased to inform you that your manuscript has been judged scientifically suitable for publication and will be formally accepted for publication once it meets all outstanding technical requirements.

Kind regards,

Dilrukshi Gamage, Ph.D

Academic Editor

PLOS ONE

Additional Editor Comments (optional):

Authors have adequately addressed the comments and improved the paper significantly. Thus I believe it is ready to be accepted.

---

## [Editor Report · Acceptance letter]

22 Jul 2024

PONE-D-24-17175R1 

PLOS ONE

Dear Dr. Kara Aydemir, 

I'm pleased to inform you that your manuscript has been deemed suitable for publication in PLOS ONE. Congratulations! Your manuscript is now being handed over to our production team.

Kind regards, 

on behalf of

Dr. Dilrukshi Gamage 

Academic Editor

PLOS ONE